# Senolytic Combination Treatment Is More Potent Than Single Drugs in Reducing Inflammatory and Senescence Burden in Cells from Painful Degenerating IVDs

**DOI:** 10.3390/biom13081257

**Published:** 2023-08-16

**Authors:** Matthew Mannarino, Oliver Wu-Martinez, Kai Sheng, Li Li, Rodrigo Navarro-Ramirez, Peter Jarzem, Jean A. Ouellet, Hosni Cherif, Lisbet Haglund

**Affiliations:** 1Department of Surgery, Orthopaedic Research Laboratory, McGill University, Montreal, QC H3G 1A4, Canada; matthew.mannarino@mail.mcgill.ca (M.M.); oliver.wumartinez@mail.mcgill.ca (O.W.-M.); kai.sheng@mail.mcgill.ca (K.S.); li.li17@mail.mcgill.ca (L.L.); peter.jarzem@mcgill.ca (P.J.); jean.a.ouellet@mcgill.ca (J.A.O.); hosni.cherif@mail.mcgill.ca (H.C.); 2Department of Surgery, McGill Scoliosis and Spine Group, McGill University, Montreal, QC H3G 1A4, Canada; neuronavarro@gmail.com; 3Shriner’s Hospital for Children, Montreal, QC H4A 0A9, Canada; 4Montreal General Hospital, 1650 Cedar Avenue, C.10.166, Montreal, QC H3G 1A4, Canada

**Keywords:** low back pain, intervertebral disc degeneration, cellular senescence, senotherapeutics, senolytics, combination therapy

## Abstract

Background: Low back pain is a global health problem directly related to intervertebral disc (IVD) degeneration. Senolytic drugs (RG-7112 and o-Vanillin) target and remove senescent cells from IVDs in vitro, improving tissue homeostasis. One drawback of using a single senolytic agent is the failure to target multiple senescent antiapoptotic pathways. This study aimed to determine if combining the two senolytic drugs, o-Vanillin and RG-7112, could more efficiently remove senescent cells and reduce the release of inflammatory factors and pain mediators in cells from degenerating human IVDs than either drug alone. Methods: Preliminary data evaluating multiple concentrations of o-Vanillin and RG-7112 led to the selection of four treatment groups. Monolayer and pellet cultures of cells from painful degenerate IVDs were exposed to TLR-2/6 agonist. They were then treated with the senolytics o-Vanillin and RG7112 alone or combined. p16*^ink4a^*, Ki-67, caspase-3, inflammatory mediators, and neuronal sprouting were assessed. Results: Compared to the single treatments, the combination of o-Vanillin and RG-7112 significantly reduced the amount of senescent IVD cells, proinflammatory cytokines, and neurotrophic factors. Moreover, both single and combination treatments significantly reduced neuronal sprouting in rat adrenal pheochromocytoma (PC-12 cells). Conclusions: Combining o-Vanillin and RG-7112 greatly enhanced the effect of either senolytic alone. Together, these results support the potential of senolytics as a promising treatment for IVD-related low back pain.

## 1. Introduction

Low back pain is experienced by approximately 80% of individuals at some point in their lifetime. Globally, it is the number one cause of years lived with disability [1]. Despite its prevalence, little is known about the cellular and molecular mechanisms leading to painful intervertebral disc (IVD) degeneration, leaving surgical removal and vertebral fusion in end-stage disease as the mainstay of treatment. This age-related health problem is associated with IVD degeneration in many individuals [2].

Healthy, painless IVDs are largely avascular and aneural. Evidence shows that degenerating, painful IVDs are innervated [3], inferring that discogenic pain is associated with increased IVD innervation. Chronic pain induced by disc degeneration is speculated to stem from nerve damage and neuronal sensitization [4,5]. Moreover, inflammatory pathways involved in IVD degeneration have been linked to chronic back pain [5]. In vitro, increases in proinflammatory cytokines (interleukins; IL-1β, IL-6, and tumor-necrosis-factor-alpha (TNF-α)) and pronociceptive factors (nerve growth factor (NGF) and brain-derived neurotrophic factor (BDNF)) have been observed in cells isolated from degenerating IVDs [6,7]. NGF and BDNF promote neuronal development, survival, and growth [3]. Moreover, it is known that degenerating human IVDs produce sufficient NGF and BDNF to promote neurite growth and nociceptive peptide production [8].

Targeting cellular senescence (i.e., viable cells which can no longer divide) is a novel venue currently being investigated to treat chronic lower back pain. To date, the literature suggests that senescent cell accumulation during tissue degeneration contributes directly to the onset and progression of degenerative musculoskeletal illnesses such as osteoarthritis [9] and IVD degeneration.

The senescence-associated secretory phenotype (SASP) refers to the secretion of cytokines, chemokines, neurotrophins, growth factors, and proteases by senescent cells [10,11]. These SASP factors are believed to promote matrix catabolism, sterile inflammation and pain in IVDs, thereby speeding up the degenerative process in a paracrine manner [12].

In human IVD, there is an expression of Toll-like receptor (TLR)-1, -2, -3, -4, -5, -6, -9, and 10, and the expression of TLR-1, -2, -4, and -6 are increased with the degree of disc degeneration and pain [13,14]. Overloading intact disc and IVD cells can upregulate TLR-2 and -4 expressions, and previous data from our lab demonstrate that activating TLR receptors with the synthetic agonists (Pam2CSK4, TLR-2/6 agonist) induced IVD degeneration, pain, and cellular senescence [13,15].

Synthetic drugs that selectively target and remove senescent cells (senolytic) have recently been identified [16,17]. Two compounds of interest are RG-7112 and o-Vanillin. The FDA-approved compound RG-7112 (RO5045337) [18] restores p53 physiological activity via MDM2 [18,19]. RG-7112 has been shown to specifically kill senescent IVD cells [15,20]. o-Vanillin is a natural senolytic compound recognized for its antioxidant and anti-inflammatory properties, which have been reported to have senotherapeutic action on IVD cells from degenerating painful human IVDs and on IVD cells where senescence was induced by activation of TLR-2 [15,20,21].

Senescent cells are heterogenous, and one drawback of using a single senolytic agent is the failure to target multiple senescent antiapoptotic pathways in the same cell type or different cell populations within a target tissue. Concurrently targeting multiple and indirectly related antiapoptotic pathways may result in increased targeting of senescent cells in the absence of toxicity for normal proliferating or quiescent cells. Successful combination therapy is exemplified by the combination of Dasatinib and Quercetin that targets antiapoptotic networks instead of a single target [22]. The lower therapeutic dosages enabled by the combination also decreased side effects associated with single drugs [22].

Creating environments that reflect the true milieu in humans is critical for translation from the bench to medical practice. Moving away from 2D monolayer cultures, which do not provide a realistic environment seen in humans (e.g., lack of matrix), is critical. This is especially important when investigating senolytics as, in reality, drugs do not overlay on cells but instead need to infiltrate through multiple regions to provide an effect, in this case, on the intervertebral disc. As such, this has led to increased use of 3D in vitro pellet or organoid cell cultures.

The current manuscript used a 3D culture model of cells from painful degenerate IVD and stimulated them with a TLR-2 agonist to heighten the senescent phenotype. The objectives were a) to determine if treatment with o-Vanillin and/or RG-7112 has the potential to alter factors that directly affect innervation, b) to determine if a combination treatment with o-Vanillin and RG-7112 was more potent than single treatment at reducing senescent phenotypes and SASP factors.

## 2. Materials and Methods

### 2.1. Tissue Collection and Cell Isolation

All procedures performed were approved by the ethical review board at McGill University (IRB # 2019-4896). Degenerate IVDs were obtained from chronic low back pain patients who received discectomies to alleviate pain. Donor information is presented in Appendix A. IVD cells were isolated, as previously described [15]. Briefly, samples were washed in phosphate-buffered saline solution (PBS, Sigma-Aldrich, Oakville, ON, Canada) and Hank’s-buffered saline solution (HBSS, Sigma-Aldrich, Oakville, ON, Canada) supplemented with Primocin^TM^ (InvivoGen, San Diego, CA, USA) and Fungiozone (Sigma-Aldrich, Oakville, ON, Canada). Then, the matrix was minced and digested in 0.15% collagenase type II (Gibco) for 16 h at 37 °C. Cells were passed through both a 100 μm filter and 70 μm filter before being resuspended in Dulbecco’s Modified Eagle Medium (DMEM, Sigma-Aldrich, Oakville, ON, Canada) supplemented with 10% fetal bovine serum (FBS, Gibco), Primocin^TM^, Glutamax (Oakville, ON, Canada), and maintained in a 5% CO_2_ incubator at 37 °C.

### 2.2. In Vitro Cell Culture and Treatment

Monolayer culture: Experiments were performed with degenerate IVDs (NP and AF cells) within passages 1 to 2. Then, 20,000 cells were seeded in 8-well chamber slides (Nunc™ Lab-Tek™ II Chamber Slide™ System) for immunocytochemistry experiments following treatment. Three hundred thousand cells were seeded in 6-well plates (Sarstedt, TC plate 6-well, Cell+, F) for ELISA and RNA extraction following treatment. All cells were left to adhere for 12 to 24 h and then serum-starved in DMEM with 1X insulin–transferrin–selenium (ITS, Thermo Fisher, Waltham, MA, USA) for 6 h prior to treatment. Induction of senescence was performed by exposing cells to 100 ng/mL of Pam2CSK4 (TLR-2/6 agonist, Invivogen) for 48 h. Cells were treated with senolytic compounds for the last 6 h of the incubation [15]. Serial dilutions with 0.25/0.5/2.5/5 μM of RG-7112 (Selleck Chemicals, Houston, TX, USA) or 5/10/50/100 μM of o-Vanillin (Sigma-Aldrich, Oakville, ON, Canada) were performed to identify concentrations that are the most potent and the lowest concentration of each senolytic. Combination treatments were as follows: 5 μM of RG-7112 and 100 μM of o-Vanillin, 2.5 μM of RG-7112 and 100 μM of o-Vanillin, 5 μM of RG-7112 and 50 μM of o-Vanillin or 2.5 μM of RG-7112 and 50 μM of o-Vanillin was performed. The concentration of both compounds is based on previously published studies from our laboratory as well as work done by other groups working with o-Vanillin and RG-7112 [15,20,21,23,24].

Pellet culture: 100,000 cells/tube were collected by centrifugation at 300 g for 5 min. Pellets were incubated in 300 μL of DMEM, 2.25 g/L of glucose (Sigma-Aldrich, Oakville, ON, Canada), 5% FBS, 5 μM of ascorbic acid, 1% GlutaMAX, 0.5% Gentamicin (Thermo Fisher, Waltham, MA, USA) at 37 °C and 5% CO_2_. Pellets were left in DMEM for two days to form and stabilize, followed by two days of induction with 100 ng/mL of Pam2CSK4 and then treated with either 100 μM of o-Vanillin, 5 μM of RG-7112, 5 μM of RG-7112 and 100 μM of o-Vanillin, 2.5 μM of RG-7112 and 100 μM of o-Vanillin, 5 μM of RG-7112 and 50 μM of o-Vanillin or 2.5 μM of RG-7112 and 50 μM of o-Vanillin for four days; meanwhile pellets in the control group stayed in DMEM (Sigma-Aldrich, Oakville, ON, Canada) (Combination treatment with o-Vanillin and RG-7112 results in additive apoptotic and proliferative activity in pellet cultures. A). Following the treatment period, pellets from all groups were cultured for 16 days, and their culture media were collected every 4 days and pooled as posttreatment media (Combination treatment with o-Vanillin and RG-7112 results in additive apoptotic and proliferative activity in pellet cultures. A). After day 24, cell pellets were washed in PBS, fixed with 4% paraformaldehyde (Thermo Fisher, Waltham, MA, USA), cryoprotected in 10–30% sucrose, and then transferred via spatula into a plastic mold for embedment in Optimum Cutting Temperature compound (OCT, Thermo Fisher, Waltham, MA, USA), and finally flash-frozen at −80 °C and kept at −20 °C. Sections were cut 5 µm thick with a cryostat (Leica Microsystems, Richmond Hill, ON, Canada) and placed on slides for immunostaining.

PC12 culture: Rat adrenal pheochromocytoma (PC12) cell line expresses the receptor for and responds to NGF. When exposed to NGF, they take on a neuronal-like phenotype. They are commonly used to study neuronal differentiation and neurite sprouting [8,25]. PC12 cells (ATCC, Manassas, VA, USA) in passages 2–7 were cultured on 6-well plates (Nunc) or 8-well chamber slides (Nunc) coated with 50 μg/mL rat tail collagen type I (Gibco) and 10 μg/mL of Poly-l-Lysine (Sigma-Aldrich). The cells were maintained for 24 h in RPMI (Gibco) media containing 10% horse serum (Gibco), 5% FBS (Gibco) and 1× antibiotic/antimycotic (anti–anti) solution (Gibco). PC12 and neuronal culture media were replaced after a 24 h acclimatization period with day 12 media from nontreated and treated pellet cultures. PC12 cells were exposed for 48 h to the different media (*n* = 5 with no senolytic treatment, *n* = 5 treatment with 5 μM of RG-7112 and 100 μM of o-Vanillin, *n* = 5 with 2.5 μM of RG-7112 and 100 μM of o-Vanillin and *n* = 5 with 5 μM of RG-7112 and 50 μM of o-Vanillin).

### 2.3. Immunofluorescence

Pellet sections were heated to 60 °C for 30 min, then washed and permeabilized with PBS 1% Tween and 0.1% Triton. Cells were blocked with 0.3% Triton X-100 in PBS, saturated with 1% BSA, 1% serum, and 0.1% Triton X-100 for 2 h and then incubated for 2 h at room temperature with p16*^ink4a^* (Roche, Ventana laboratories, Mississauga, ON, Canada), Ki-67 (Novus, Oakville, ON, Canada) and caspase-3 (Sigma-Aldrich, Oakville, ON, Canada) primary antibodies. After washing, cells were incubated with the appropriate Alexa Fluor^®^ 488- or 555-conjugated secondary antibody (Thermo Fisher, Waltham, MA, USA) for 2 h at room temperature and then counterstained with DAPI for nuclear staining. Photomicrographs were acquired with a fluorescent Olympus BX51 microscope equipped with an Olympus DP71 digital camera (Olympus, Tokyo, Japan). Over 1000 cells were quantified for each pellet. Positive cell percentage was quantified by Fiji ImageJ (version: 2.1.0/1.53c). Briefly, the number of cells stained positive for one of the target proteins (p16^INK4a^, Ki-67 or caspase-3) was counted and compared to the total number of cells positive for DAPI staining.

### 2.4. PC12 Culture Image Acquisition and Neurite Analysis

Each medium was applied in duplicate wells, and two random images per well were taken using a Zeiss Axiovert 40 C inverted light microscope (Toronto, ON, Canada) with a Canon PowerShot A640 camera and 52 mm Soligor adaptor tube (Mississauga, ON, Canada). The graph is generated by counting 10 images from 7 different donors per group. This would equal over 2000 cells being quantified for each condition. The percentage of cells with neurites was determined and then averaged for each experimental condition.

### 2.5. Immunocytochemistry

Monolayer cultures were washed with PBS, fixed with 4% paraformaldehyde (Thermo Fisher, Waltham, MA, USA), and blocked in PBS with 1% BSA (Sigma-Aldrich, Oakville, ON, Canada), 1% serum, and 0.1% Triton X-100 (Sigma-Aldrich, Oakville, ON, Canada) for 1 h. Slides were then incubated with the p16*^ink4a^* CINtec PLUS Kit (Roche, Ventana laboratories, Mississauga, ON, Canada), according to the manufacturer’s instructions. Slides were also exposed to primary antibodies specific to Ki-67 (Novus, Oakville, ON, Canada) and caspase 3 (Sigma-Aldrich, Oakville, ON, Canada) overnight at 4 °C. A mouse- and rabbit-specific HRP/DAB (ABC) Detection IHC Kit (ab64264, Abcam, Cambridge, Ma, USA) was used for p16*^ink4a^*, caspase-3 and Ki-67 staining. Images were processed using AxioVision LE64 software 4.9.1(Zeiss, Oberkochen, Germany). Ten fields, randomly distributed across the well, were analyzed, total cells were counted, and the percentage of positive cells was calculated.

### 2.6. Real-Time Quantitative Polymerase Chain Reaction (RT-qPCR)

RNA was extracted using the TRIzol chloroform extraction method previously described for surgical samples and PC12 cells [8]. Then, 500 ng of RNA were reverse-transcribed using a qScript cDNA Synthesis Kit (Quanta Biosciences, Beverly, MA, USA) with an Applied Biosystems Verti Thermocycler (Thermo Fisher, Waltham, MA, USA). RT-qPCR was performed using an Applied Biosystems StepOnePlus machine (Thermo Fisher, Waltham, MA, USA) with PerfecCTa SYBR Green Fast Mix (Quanta Biosciences, Beverly, MA, USA). Primer sequences for TLRs, senescent markers, neurite growth markers, as well as pain and inflammatory markers (p16^ink4a^, TNF-α, CXCL-10, CXCL-1, CXCL-8, GM-CSF, TGF-β, CCL-2, CCL-5, CCL-7, CCL-8, NGF, BDNF, TLR-1, -2, -4, -6, neurofilament light chain (NF-L), plasminogen activator urokinase receptor (Plaur), Polo-like kinase 2 (Plk2), poliovirus receptor (PVR), vaccinia growth factor (VGF)) and the housekeeping gene (GAPDH), can be found in Appendix A. All reactions were conducted in technical triplicate, and fold changes in gene expression were calculated using the 2^−ΔΔCt^ method after normalizing to GAPDH and nontreated samples [26].

### 2.7. ELISA

Cell culture media from degenerate IVD cells cultured in monolayer and pellets were used to assess the concentrations of IL-6, IL-8, IL-1β, TNF-α, NGF and BDNF. Next, 150 μL of monolayer culture media and day 12 pellet culture media were used. ELISAs were performed as per the manufacturer’s instructions (RayBiotech, Norcoss, GA, USA). Colorimetric absorbance was measured with a Tecan Infinite M200 PRO (Tecan, Männedorf, Switzerland) spectrophotometer and analyzed with i-control 1.9 Magellan software 1.68(Tecan, Männedorf, Switzerland). Protein levels of the treated conditions and controls were then compared.

### 2.8. Dimethylmethylene Blue Assay

Dimethylmethylene Blue (DMMB) assays were conducted as previously described [23] to quantify sulfated glycosaminoglycans (sGAG) in the day 12 IVD pellet media. Chondroitin sulfate was used to generate the standard curve. All samples were ensured to fall into the linear portion of the standard curve. Each sample was triplicated into clear 96-well plates (Costar, Corning, NY, USA). DMMB dye was then added to the wells. The absorbance was measured immediately at room temperature using a Tecan Infinite T200 spectrophotometer (Männedorf, Switzerland).

### 2.9. Metabolic Activity

Metabolic activity was assessed by the Alamar Blue assay [23]. Briefly, on day 24, surgical sample cell pellets were exposed to 10% Alamar Blue reagent (Thermo Fisher, Waltham, MA, USA) in DMEM and incubated for 18 h at 37 °C. Fluorescence (Ex560/Em590) was measured by using a spectrophotometer (Tecan Infinite T200, Männedorf, Switzerland) equipped with Magellan software (Tecan, Männedorf, Switzerland). The results are presented as a percentage of metabolic activity compared to the control. Experiments were performed in triplicate wells for each pellet condition.

### 2.10. Statistical Analysis

Data were analyzed using Graph Prism 9 (GraphPad, La Jolla, CA, USA). Analysis was performed using the paired Student’s *t*-test or one-way ANOVA. Specific tests are indicated in the figure legends with the corrections. A *p*-value <0.05 was considered statistically significant. Data are presented as mean ± SD.

## 3. Results

### 3.1. RG-7112 Reduced the Expression of p16^ink4a^, TLR-2, and SASP Factors in IVD Cells from Patients with Back Pain and IVD Degeneration

IVD cells from patients with low back pain and IVD degeneration were exposed to a TLR-2/6 agonist for 48 h in the presence or absence of RG-7112 (5 μM) during the last 6 h. We have previously observed that exposure of IVD cells from patients with low back pain and IVD degeneration to a TLR-2/6 agonist significantly increases the expression of SASP factors, *p16^ink4a^* and TLR-2 [15]. We have also reported that o-Vanillin, a natural senolytic, can reduce senescence and SASP factors in this TLR-2/6-induced senescent human IVD cells [15]. Here, we sought to investigate if RG-7112 has a similar impact [15]. Like o-Vanillin, when compared to the control, RG-7112 significantly reduced the expression of CCL2 from 42.32 ± 11.34 to 2.3 ± 3.29 (*p* = 0.004860), CCL5 from 49.03 ± 11.49 to 15.00 ± 6.14 (*p* = 0.015869), CCL7 from 9.30 ± 1.43 to 4.96 ± 2.20 (*p* = 0.015001), CCL8 from 28.40 ± 4.94 to 2.91 ± 2.40 (*p* = 0.003127), GM-CSF from 118.55 ± 10.07 to 7.15 ± 4.83 (*p* = 0.000239), BDNF from 1.77 ± 0.13 to 1.07 ± 0.16 (*p* = 0.002507), NGF from 1.35 ± 0.11 to 0.61 ± 0.19 (*p* = 0.000954), TNF-α from 7.36 ± 2.36 to 1.77 ± 0.77 (*p* = 0.022738), CXCL1 from 594.16 ± 44.72 to 38.66 ± 36.54 (*p* = 0.000002), CXCL8 from 594.50 ± 98.64 to 209.32 ± 157.89 (*p* = 0.022738), and CXCL10 from 745.23 ± 107.79 to 74.22 ± 89.64 (*p* = 0.001659) (Figure 1A). In addition, treatment with RG-7112 significantly decreased p16*^ink4a^* expression from 3.83 ± 1.05 in the control to 1.21 ± 0.22 in RG-7112 (*p* = 0.004324) (Figure 1B). Lastly, a significant decrease in TLR-2 expression (*p* = 0.009338) was also observed in RG-7112 (2.69 ± 0.98)-treated cells compared with the control (9.17 ± 1.29), as has previously been reported for o-Vanillin [15] (Figure 1C).

Additionally, SASP factor release into the culture media was assessed from cultures exposed to a TLR-2/6 agonist for 48 h in the presence or absence of RG-7112 (5 μM) during the last 6 h of culture. As previously reported [15], treatment with TLR-2/6 agonist Pam2CSK4 resulted in a significant increase in SASP factor expression (namely TNF-α, IL-1β, IL-8 and NGF) when compared to the control. Interestingly, this induction was significantly decreased for all evaluated SASP factors following treatment with RG-7112. TNF-α expression decreased from 289.67 pg/mL ± 25.69 in the control to 99.99 pg/mL ± 19.91, *p* = 0.000068 in RG-7112-treated cells, IL-1β expression decreased from 40.21 ± 3.53 pg/mL in the control to 20.68 pg/mL ± 3.86, *p* = 0.000098 in RG-7112-treated cells, IL-8 expression decreased from 410.01 pg/mL ± 30.47 in the control to 147.92 pg/mL ± 16.56, *p* = 0.000030 in RG-7112-treated cells and NGF expression decreased from 358.43 pg/mL ± 19.85 in the control to 262.32 pg/mL ± 6.96, *p* = 0.000227 in RG-7112-treated cells (Figure 1D).

### 3.2. Identifying the Lowest Effective Concentration of RG-7112 and o-Vanillin at Which Senolytic Activity Is Preserved in Monolayer Culture

Given the role of o-Vanillin in TLR-2/6-induced senescence and the findings above, which demonstrate a similar role to RG-7112, we investigated whether there is an additive effect of combining the two senolytic compounds. First, to determine the lowest concentration with senolytic activity, IVD cells from patients with low back pain and IVD degeneration were exposed to the TLR-2/6 agonist for 48 h in the presence of serial dilutions of either RG-7112 or o-Vanillin during the last 6 h of the treatment. Control samples only received induction with the TLR-2/6 agonist. A significant decrease in senescent cells (i.e., p16*^ink4a^*-positive cells, nuclear localization) was observed with all four dilutions of RG-7112 (0.25 μM, 0.5 μM, 2.5 μM, 5 μM) and three dilutions of o-Vanillin (10 μM, 50 μM, 100 μM). Representative images can be seen in Figure 2A(a,d). Relative to the control (21.24% ± 1.23), the percentage of p16*^ink4a^*-positive senescent cells with RG-7112 was 18.50% ± 1.61 at 0.25 μM (*p* = 0.0437), 14.58% ± 0.42 at 0.5 μM (*p* = 0.004), 13.25% ± 0.46 at 2.5 μM (*p* < 0.0001), and 10.57% ± 0.48 at 5 μM (*p* < 0.0001). With o-Vanillin, the remaining percentage of p16*^ink4a^*-positive cells was 15.59% ± 1.42 at 10 μM (*p* = 0.0076), 14.21% ± 1.41 at 50 μM (*p* = 0.0013) and 11.17% ± 1.01 at 100 μM (*p* < 0.0001) (Figure 2B). The number of Ki-67-positive proliferative cells was significantly (*p* = 0.0149) increased to 21.50% ± 0.75 following treatment with 5 μM RG-7112 treatment and to 21.61% ± 1.22 (*p* = 0.0093) following 100 μM o-Vanillin treatment as compared to the untreated control (18.36% ± 0.96) (Figure 2A(b,e),C). Both compounds also increased the number of caspase-3-positive apoptotic cells (Figure 2A(c,f),D). Caspase-3-positive cells increased to 11.74% ± 0.86 (*p* = 0.0141) for 5 μM RG-7112 and to 10.87% ± 1.03 (*p* = 0.0144) for 100 μM o-Vanillin compared with the untreated control (9.09% ± 0.77).

Given that 5 μM of RG-7112 and 100 μM of o-Vanillin resulted in the most significant decrease in senescence, combinations of these concentrations were selected for future experiments. Moreover, 2.5 μM of RG-7112 and 50 μM of o-Vanillin were further investigated, given that they both resulted in a significant decrease in the expression of p16*^ink4a^*. Three combinations were assessed: 5 μM of RG-7112 and 100 μM of o-Vanillin; 2.5 μM of RG-7112 and 100 μM of o-Vanillin; and 5 μM of RG-7112 and 50 μM of o-Vanillin.

### 3.3. A Combination of o-Vanillin and RG-7112 Has Additive Apoptotic and Proliferative Activity in Pellet Cultures

To further assess the combination of RG-7112 and o-Vanillin, pellet cultures of IVD cells from patients with low back pain and IVD degeneration were exposed to the TLR-2/6 agonist for 48 h, treated with senolytics (in combination or single drugs) for four days and then cultured for a total of 24 days (Figure 3A). The control group only received TLR-2/6 agonist for 48 h but no senolytics. Pellet cultures of IVD cells were used as they have been suggested to mimic three-dimensional in vivo conditions more closely, allowing for the evaluation of the matrix synthesis.

The pellet cultures’ expression of p16*^ink4a^*, apoptotic (cleaved caspase-3, nuclear localization), and proliferative (Ki-67, nuclear localization) capacities were assessed. All treatments significantly reduced the number of senescent cells compared to the control. The largest decrease in *p16^ink4a^*-positive cells was observed in pellet cultures treated with the combination of 5 μM of RG-7112 and 100 μM of o-Vanillin. Control cultures had 18.97% ± 1.89 while treated with combination cultures had 3.74% ± 1.52 (*p* = 0.0004) senescent cells (Figure 3B(b,g),D). The second-most efficient treatment was with a combination of 2.5 μM of RG-7112 and 100 μM of o-Vanillin, which reduced senescent cells to 5.77% ± 1.38 (*p* = 0.0002), followed by 5 μM of RG-7112 and 50 μM of o-Vanillin, which reduced senescent cells to 9.26 ± 1.35 (*p* = 0.0092) and 5 μM of RG-7112 and 100 μM of o-Vanillin applied as single drugs, which reduced the number of senescent cells to 11.09% ± 0.98 (*p* < 0.0001), and 10.31% ± 0.84 (*p* < 0.0001), respectively (Figure 3D). Compared to RG-7112 and o-Vanillin alone, only 5 μM of RG-7112 and 100 μM of o-Vanillin (relative to RG-7112 *p* = 0066, relative to o-Vanillin *p* = 0.0039) and 2.5 μM of RG-7112 and 100 μM of o-Vanillin (relative to RG-7112 *p* = 0.0033, relative to o-Vanillin *p* = 0.0071) significantly enhance the effect.

The effect on proliferation with the greatest increase in Ki-67-positive cells was observed when treating cells with 5 μM of RG-7112 and 100 μM of o-Vanillin relative to the control. Proliferating cells increased from 15.24% ± 1.10 to 36.74% ± 3.25 (*p* = 0.0011). Treatment with 2.5 μM of RG-7112 and 100 μM of o-Vanillin showed 29.27% ± 2.35 proliferating cells, and 5 μM of RG-7112 and 50 μM of o-Vanillin showed 20.93 ± 2.99 proliferating cells (*p* = 0.0415). The percentage of proliferating cells was 22.52% ± 3.08 (*p* = 0.0132) and 21.40% ± 4.17 (*p* = 0.0588) in RG-7112- and o-Vanillin-treated cultures. Compared to RG-7112 and o-Vanillin alone, only the two combinations of 100 μM of o-Vanillin with either 5 μM (relative to RG-7112 *p* = 0.0241, relative to o-Vanillin *p* = 0.0026) or 2.5 μM (relative to RG-7112 *p* = 0.0055, relative to o-Vanillin *p* = 0.0257) of RG-7112 significantly enhance the effect (Figure 3C(b,e),E). Moreover, relative to the control (10.80% ± 1.43), the greatest increase in caspase-3-positive cells was observed when treating cells with a combination of 5 μM of RG-7112 and 100 μM of o-Vanillin (17.73% ± 1.54, *p* = 0.0057) (Figure 3B(c,h),F). Interestingly, when assessing colocalization of caspase-3 and p16*^ink4a^*, combination-treated pellets (i.e., 5 µM of RG-7112 and 100 µM of o-Vanillin) had a significantly greater number of senescent cells undergoing apoptosis relative to the RG-7112 and o-Vanillin single senolytic groups (the percentage of caspase-3- and p16*^ink4a^*-positive cells colocalized in combination treatment was 54.57% ± 5.85 compared with RG-7112 35.96% ± 7.71 (*p* = 0.036) and o-Vanillin 38.04% ± 7.52 (*p* = 0.0357) (Figure 3B(d,i),G).

Furthermore, an Alamar Blue assay, which assesses metabolic activity, was performed to confirm that the RG-7112 and o-Vanillin combination treatment was not toxic to nonsenescent cells in pellet cultures from degenerate tissue. No significant difference in metabolic activity was observed in any combination or single senolytic treatment groups (Figure 3H). Finally, to assess the effect of combination treatment on matrix synthesis, a DMMB assay was performed to measure sGAG release into the culture media. An increase in proteoglycan content was observed through a significant increase in sGAG release on day 12 of culture (Figure 3I). The combination of 5 μM of RG-7112 and 100 μM of o-Vanillin resulted in the greatest sGAG release relative to the control (combination 9.74 µg/mL ± 1.17 vs. control 3.14 µg/mL ± 0.61, *p* = 0.0002) (Figure 3I). Similarly, when compared to the single treatment of RG-7112 (combination 9.74 µg/mL ± 1.17 vs. RG-7112 7.68 µg/mL ± 0.68, *p* = 0.0231) and the single treatment with o-Vanillin (combination 9.74 µg/mL ± 1.17 vs. o-Vanillin 6.89 µg/mL ± 1.27, *p* = 0.0139), the combined treatment still had higher sGAG released into the culture media (Figure 3I).

### 3.4. Combination Treatment with RG-7112 and o-Vanillin Results in a Significant Decrease of SASP Factor Release in Pellet Cultures

As has been shown, both RG-7112 and o-Vanillin each reduce SASP factor release in monolayer cultures [15] (Figure 1). Next was to evaluate whether combined treatment with these two senolytics has an additive effect on reducing SASP factor release. Pellet culture media were used to assess protein concentrations of cytokines (IL-6, IL-8, IL-1β, TNF-α) and neurotrophins (BDNF and NGF). The concentrations in the control media were IL-6: 832.9 pg/mL ± 50.39, IL-8: 639.7 pg/mL ± 40.46, IL-1β: 120.9 pg/mL ± 12.02, TNF-α: 268.8 pg/mL ± 41.67, BDNF: 1777 pg/mL ± 138.1 and NGF: 311.1 pg/mL ± 23.57. The greatest reduction in cytokine and neurotrophin expression was observed using the combination of 5 μM of RG-7112 and 100 μM of o-Vanillin. IL-6 decreased to 558.3 pg/mL ± 70.30 (*p* = 0.0034), IL-8 to 400.2 pg/mL ± 31.14 (*p* = 0.0003), IL-1β to 70.86 pg/mL ± 4.399 (*p* = 0.0002), TNF-α to 157.7 pg/mL ± 19.47 (*p* = 0.0109), BDNF to 808.5 pg/mL ± 62.77 (*p* < 0.0001) and NGF to 166.1 pg/mL ± 8.89 (*p* = 0.0001) (Figure 4A–F). However, the combination treatment was only significantly better at reducing IL-6, IL-8, and IL-1β relative to the single treatment groups (Figure 4A–C). TNF-α protein expression reduction was only significantly reduced when comparing combination treatment to single treatment with RG-7112 and not with o-Vanillin (Figure 4D). Relative to the single treatments, only the combination of 5 μM of RG-7112 and 100 μM of o-Vanillin significantly reduced BDNF (relative to RG-7112 *p* = 0.0002, relative to o-Vanillin *p* = 0.0013) and NGF protein expression (relative to RG-7112 *p* = 0.0073, relative to o-Vanillin *p* = 0.0083) (Figure 4E,F).

### 3.5. Pellet Media from RG-7112- and o-Vanillin-Treated Pellets Induced Less Neurite Sprouting and Neurite Growth Gene Expression in PC-12 Cells

As seen from the ELISA data above, two of the SASP factors released after induction with TLR-2/6 agonist in cells from painful degenerate IVDs are NGF and BDNF. It has previously been shown that stimulation with NGF can cause neurite sprouting in PC-12 cells. Therefore, to assess the difference in levels of PC-12 sprouting, these cells were cultured for 48 h in culture media of nontreated and senolytic-treated pellets. Gene expression analysis of common neuronal markers was performed to confirm that pellet culture media induce neuronal differentiation. After 48 h of culture, NF-L, Plaur, Plk2, PVR, and VGF were significantly lower in the 5 μM of RG-7112 and 100 μM of o-Vanillin combination-treated culture media samples than that of the nontreated control (control: NF-L: 12.06 ± 1.65, Plaur: 6.21 ± 0.39, Plk2: 12.40 ± 0.73, PVR: 11.88 ± 1.46 and VGF: 6.00 ± 0.48) and (combination: NF-L: 4.54 ± 0.79, *p* = 0.001669; Plaur: 2.10 ± 0.61, *p* = 0.001669; Plk2: 5.05 ± 0.57, *p* = 0.002624; PVR: 5.88 ± 0.42, *p* = 0.001669; VGF: 2.07 ± 0.72, *p* = 0.001669) (Figure 5A). Interestingly, for NF-L, Plaur and PVR, the combination treatment (5 μM of RG-7112 and 100 μM of o-Vanillin) was only better than the RG-7112 treatment groups (NF-L: RG-7112: 7.8 ± 1.08, *p* = 0.008804; Plaur: RG-7112: 3.39 ± 0.39, *p* = 0.003223; PVR: RG-7112: 8.63 ± 1.30, *p* = 0.002073) (Figure 5A). Next was to observe the number of PC-12 cells that had neurite growth. The lowest number of neurite growth was found in the 5 μM of RG-7112 and 100 μM of o-Vanillin combination-treated group when compared to the nontreated group (combination 60.93% ± 4.00 vs. control 39.91% ± 2.92, *p* < 0.0001) (Figure 5B,C and Appendix A). When compared to the single-treatment groups, the combined treatment (5 μM of RG-7112 and 100 μM of o-Vanillin) had significantly decreased the number of cells with extended neurites from 60.93% ± 4.00 for the combination to 49.86% ± 3.84 for RG-7112 (*p* < 0.0001), and to 47.74% ± 4.65 (*p* < 0.0001) for o-Vanillin (Figure 5B,C and Appendix A).

## 4. Discussions

Previously, we demonstrated that TLR-2/6 activation induces cell senescence and SASP factor release in IVD cells [15]. We further illustrated that using o-Vanillin, a senolytic compound, senescence was attenuated in TLR-2/6-induced IVD cells [15]. Our lab has also shown that the senolytic compound RG-7112 has senotherapeutic activity on IVD cells [20]. The role of RG-7112 is further elucidated in this manuscript, where we demonstrated that, like o-Vanillin, RG-7112 could attenuate the effect of TLR-2/6 induction on cells from painful degenerate IVD. Since both compounds have similar effects but target different pathways, there is a potential for these compounds to work when used in combination. Therefore, one objective of the current study was to demonstrate that combining o-Vanillin and RG-7112 can supersede the effect of o-Vanillin or RG-7112 alone. Moreover, knowing that cells from painful degenerate IVD can drive neurite sprouting [8], we sought to determine if treating IVD cells with senolytic compounds alone or in combination can reduce neuronal sprouting, which has been reported as an indirect indicator for chronic pain.

Cellular senescence can be induced by replicative senescence or stress-induced premature senescence [27]. In low back pain patients, cellular senescence is thought to be majorly stress-induced. Using a model with a TLR-2/6 agonist, we can mimic a “stress-induced” environment often seen in degenerate IVDs [15]. Eliminating senescent cells from multiple tissues or even a single tissue will probably require combinations of multiple senotherapeutic drugs [22]. This model shows that 100 μM of o-Vanillin or 5 μM of RG-7112 each provide senolytic activity. One benefit of combining treatments is that each independent compound’s concentration could be reduced, reducing the risk of side effects. Previous studies have demonstrated that lower concentrations of RG-7112 such as those used here are safe and beneficial in musculoskeletal diseases, whereas, at the high concentrations, used for hematological cancer therapy, toxic side effects were described [28]. As for o-Vanillin, the safety of the compound was evaluated to have an IC50 = 4.2 mM, but no higher concentration than 100 μM was tested. To find the concentrations at which each drug maintained senolytic potential, dilutions of o-Vanillin and RG-7112 were evaluated in monolayer cultures. The senolytic capabilities (i.e., remove p16*^ink4a^*-positive cells, increase proliferation and increase apoptosis) were reduced once the concentration of o-Vanillin is below 50 μM and the concentration of RG-7112 is below 2.5 μM. Being that this is the first time that the combination of o-Vanillin and RG-7112 is being assessed, we decided only to use concentrations at which the senolytic compounds alone demonstrated senolytic activity, which is why we did not go below 50 μM for o-Vanillin and 2.5 μM for RG-7112 in the treatment groups (single or combined treatment).

The assessment of monolayer culture is an excellent model to determine if compounds have senolytic effects; however, to better understand if there is any physiological relevance to these treatments, a model that mimics the physiological environment of the human disc is required [20,23,29]. As previously described, the 3D pellet model is a relevant model to simulate the physiological environment and matrix formation of the IVDs [20,23,29]. In this model, we induced senescence with a TLR-2/6 agonist and treated the cells with senolytics. The combination of 100 μM of o-Vanillin + 5 μM of RG-7112 had a greater effect than the single treatment in reducing senescence and SASP factor release. It also increased senescent cells undergoing apoptosis and the number of proliferating nonsenescent cells. Proteoglycan production was also enhanced. These data concord with published data [30,31]. This potential additive effect between o-Vanillin and RG-7112 is similar to that previously published in the literature for combination therapy with Fisetin, Dasatinib and Quercetin in the context of osteoarthritis [22]. Like o-Vanillin and RG-7112, Dasatinib and Quercetin target antiapoptotic networks instead of a single target. This was possible by leveraging natural and synthetic forms of senolytics. Moreover, with Dasatinib and Quercetin combination treatment, lower therapeutic dosages were used, and fewer side effects were observed than with the single-treatment groups. Recently, it has been shown that multiple and long-term injections of a combination of Dasatinib and Quercetin in elderly mice reduce p16*^ink4a^* and SASP expression levels in the IVD [30]. Dasatinib and Quercetin combination treatment also increased the expression of extracellular matrix proteins, thereby restoring tissue homeostasis [30]. A phase II clinical trial initiated in 2020 is currently undergoing using a combination of the senolytic compounds Fisetin, Dasatinib, and Quercetin to target cellular senescence to improve skeletal health in older humans; the data are set to be available in 2023 [22]. Of note, it is difficult to compare the combination treatment of Dasatinib and Quercetin with the RG-7112 and o-Vanillin combination, since the Dasatinib and Quercetin study did not provide a single-drug control [30]. Collectively, this supports the need to investigate further the role of senolytics in managing cellular medical conditions.

With the growing body of evidence supporting the critical value of senolytic compounds in musculoskeletal diseases, there has been an increasing interest in evaluating the effect of senolytic compounds in chronic pain. An association between chronic pain and senescence has previously been suggested in mice subjected to experimental nerve damage. Here, reduced telomere length and p53-mediated cellular senescence in the spinal cord resulted in mice maintaining a high pain level [32]. Low back pain is often associated with IVD degeneration, but how disc degeneration is related to low back pain is not fully understood. Pain is thought to develop through a complex interplay between IVD matrix remodeling, the production of inflammatory, nociceptive and neurotrophic factors, nerve root and sensory neuron compression and disc innervation. Among the factors identified in IVD-conditioned media, degenerating and painful IVDs show elevated levels of NGF and BDNF [8,13]. Surprisingly, the relatively low concentrations of these factors in the conditioned media significantly induced sprouting in the PC-12 cells [8,13]. In the literature, anti-NGF treatment has been shown to reduce neurite sprouting, demonstrating an important role for NGF in degenerating IVDs, even at low concentrations [8,13]. In the current study, combination and single treatment with o-Vanillin and/or RG-7112 significantly reduced both NGF and BDNF release compared to the untreated IVD cell pellets. Furthermore, we observed that the culture media from the treated pellets induced significantly less gene expression of markers associated with neurite sprouting and less neurite sprouting visualized in PC-12 cells. These observations are the first inference that senolytic treatment is linked to reducing pain markers in low back pain. Previously, the reduction of pain markers with a combination of senolytic drugs has been observed in an in vivo osteoarthritis model, where they observed a decrease in NGF, calcitonin gene-related peptide (CGRP) and selectively removed senescent cells [33], further supporting our findings. Overall, we hypothesize that due to the removal of senescent cells, cells secreting neurotrophic factors are reduced, which in turn limits neurite sprouting, ultimately decreasing low back pain.

Future studies should use in vivo models of disc degeneration to better represent the physiological environment of the IVD and to utilize pain behavior techniques to better understand if o-Vanillin and RG-7112 can lower pain levels in a live animal model. Moreover, a greater understanding of the molecular mechanisms of how o-Vanillin and RG-7112 work in combination for IVD degeneration may lead to improved therapies and quality of life for individuals with discogenic pain.

In conclusion, the present study shows that combining o-Vanillin and RG-7112 greatly enhanced the effect of either drug alone in reducing the number of senescent cells. Additionally, we demonstrated that combination treatment reduced gene expression of markers involved in neurite sprouting and decreased sprouting in PC-12 cells. Our results support the potential of senolytics to reduce chronic pain.

## Figures and Tables

**Figure 1 biomolecules-13-01257-f001:**
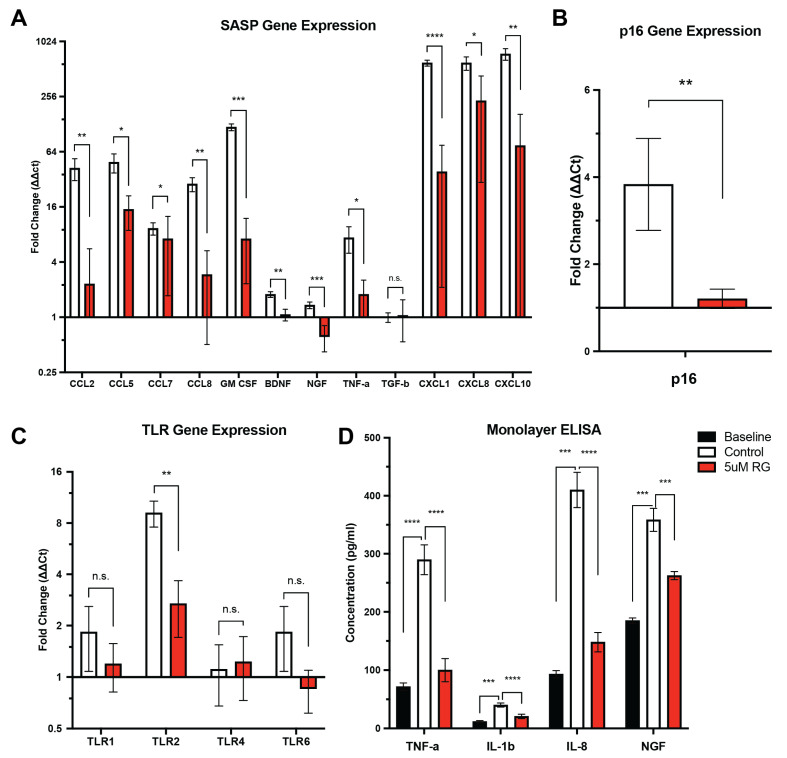
RG-7112 reduces the expression of p16, TLR-2, and SASP factors following TLR-2 activation in IVD cells from patients with back pain and IVD degeneration. (**A**–**D**) Human IVD cells from painful degenerate IVDs in monolayer culture induced with Pam2CSK4 and treated with 5 µM RG-7112. qRT-PCRs were normalized to the baseline (cells from painful degenerate IVD not induced with Pam2CSK4 and not treated with RG-7112). (**A**) Gene expression of SASP factors (CCL2, CCL5, CCL7, CCL8, GM CSF, BDNF, NGF, TNF-α, TGF-β, CXCL1, CXCL8, and CXCL10). (**B**) Gene expression of senescence marker p16. (**C**) Gene expression of Toll-like receptors (TLR-1, TLR-2, TLR-4, and TLR-6). (**D**) All monolayer culture media were analyzed by Raybio Human Cytokine Array. TNF-α, IL-1β, IL-8 and NGF protein concentration were evaluated. Values are presented as fold change ± SD in (**A**–**C**) and mean ± SD in (**D**).* Indicates significance calculated using a one-way ANOVA. * *p* < 0.05, ** *p* < 0.01, *** *p* < 0.001, **** *p* < 0.0001. (**A**–**D**) *n* = 5 donor samples per condition.

**Figure 2 biomolecules-13-01257-f002:**
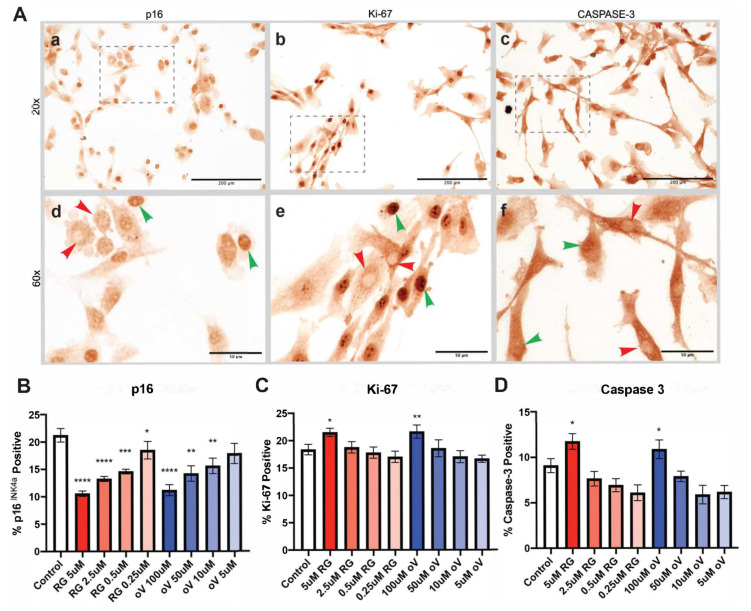
Identifying the lowest effective concentration of RG-7112 and o-Vanillin at which senolytic activity is preserved in monolayer culture. (**A**) Representative images of monolayer cultures stained with (**a**) p16, (**b**) Ki-67 and (**c**) caspase-3. (**d**–**f**) Magnified images of (**a**–**c**) with arrowheads representing positive (green) and negative (red) stained cells. Scale bars (**A**): (**a**–**c**) 200 µm and (**d**–**f**) 50 µm. Quantification of senolytic serial dilution experiments of (**B**) p16, (**C**) Ki-67 and (**D**) caspase-3 expression. Values are presented as mean ± SD in (**B**–**D**). * Indicates significance between treatment and control groups. All analyses were performed using a one-way ANOVA, *n* = 7 donor samples. * *p* < 0.05, ** *p* < 0.01, *** *p* < 0.001, **** *p* < 0.0001.

**Figure 3 biomolecules-13-01257-f003:**
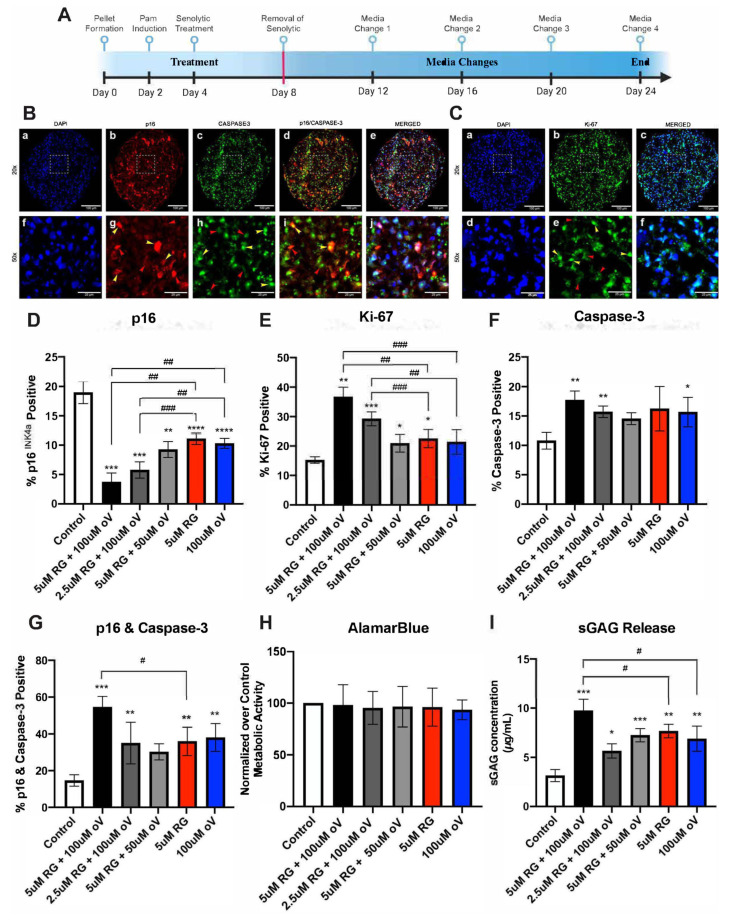
Combination treatment with o-Vanillin and RG-7112 results in additive apoptotic and proliferative activity in pellet cultures. (**A**) Timeline of pellet culture. (**B**) Representative images of pellets stained with (**a**) DAPI, (**b**) p16, (**c**) caspase-**3**, (**d**) p16 and caspase-3 merged staining, (**e**) p16, caspase-3 and DAPI merged staining. (**f**–**j**) Magnified images of (**a**–**e**) with arrowheads representing positive (yellow) and negative (red) stained cells. Scale bars (**a**–**e**) 100 µm and (**f**–**j**) 25 µm. (**C**) Representative images of pellet cultures stained with (**a**) DAPI, (**b**) Ki-67, and (**c**) DAPI and Ki-67 merged. (**d**–**f**) Represent magnified images of (**a**–**c**) with arrowheads representing positive (yellow) and negative (red) stained cells. Scale bars (**a**–**c**) 100 µm and (**d**–**f**) 25 µm. Quantification of pellet fluorescent staining of (**D**) p16, (**E**) Ki-67, (**F**) caspase-3 and (**G**) p16 and caspase-3 expression. (**H**) AlamarBlue assessed the evaluation of cell viability. (**I**) sGAG concentration measured by DMMB assay using day 12 pellet culture media. Values are presented as mean ± SD in (**D**–**I**). * Indicates significance between treatment (single or combination) and control groups. # Indicates significance calculated when combination treatment is compared to single treatment groups. All analyses were performed using a one-way ANOVA, *n* = 7 donor samples. * or # *p* < 0.05, ** or ## *p* < 0.01, *** or ### *p* < 0.001, **** *p* < 0.0001.

**Figure 4 biomolecules-13-01257-f004:**
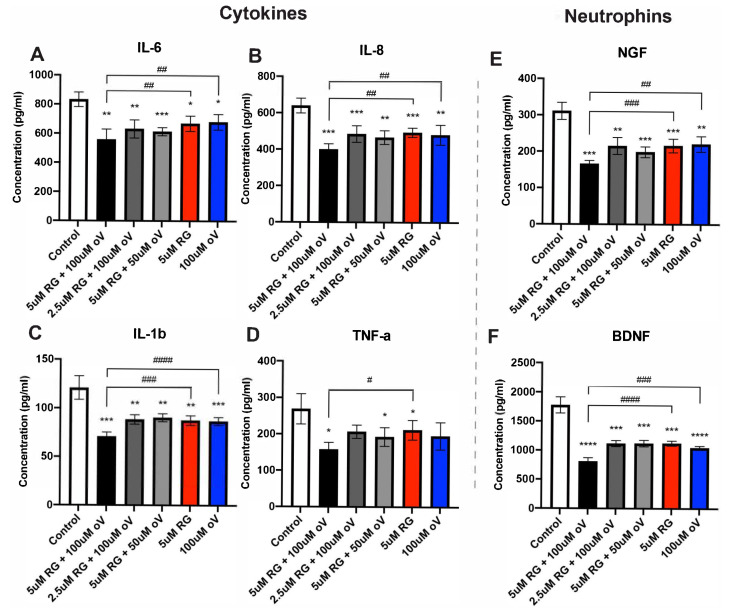
Combination treatment with RG-7112 and o-Vanillin significantly decreases SASP factor release in pellet cultures. Pellet culture media were analyzed by Raybio Human Cytokine Array. The cytokine concentration of (**A**) IL-6, (**B**) IL-8, (**C**) IL-1β, (**D**) TNF-α and neurotrophic factors (**E**) NGF and (**F**) BDNF were evaluated. Values are presented as mean ± SD in (**A**–**F**). * Indicates significance between treatment (single or combination) and control groups. # Indicates significance calculated when combination treatment is compared to single-treatment groups. All analyses were performed using a one-way ANOVA, *n* = 7 donor samples. * or # *p* < 0.05, ** or ## *p* < 0.01, *** or ### *p* < 0.001, **** or #### *p* < 0.0001.

**Figure 5 biomolecules-13-01257-f005:**
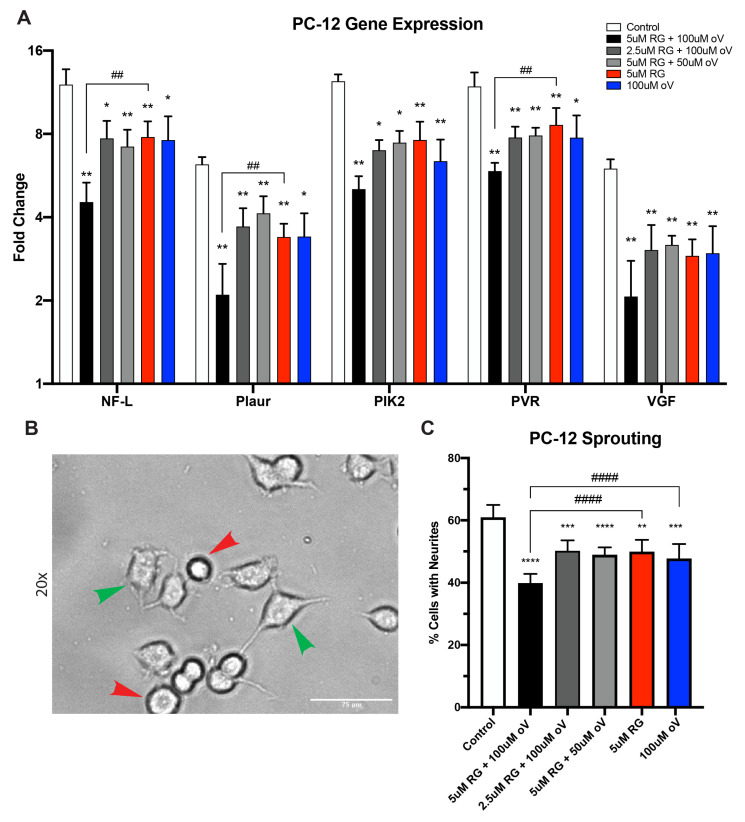
Pellet media from RG-7112 and o-Vanillin combined treated pellets have lower levels of neurite sprouting and neurite growth gene expression in PC12 cells. (**A**) Gene expression of neurite growth factors (NF-L, Plaur, PIK-2, PVR and VGF) was evaluated in PC-12 cells cultured in day 12 pellet media for 48 h. qRT-PCRs were normalized to culture media from pellets that were not induced with Pam2CSK4 and no senolytic treatment. (**B**) Representative images of neurite sprouting with arrowheads indicating neurite sprouting (green) and no neurite sprouting (red). Scale bar 75 µm. (**C**) Quantification of PC-12 neurite sprouting. Values are presented as (**A**) fold change ± SD or (**C**) mean ± SD. All analyses were performed using a one-way ANOVA, *n* = 7 donor samples. * *p* < 0.05, ** or ## *p* < 0.01, *** *p* < 0.001, **** or #### *p* < 0.0001.

## Data Availability

The data supporting this study’s findings are available from the corresponding author upon reasonable request.

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
