# Peer review of "Senolytic Combination Treatment Is More Potent Than Single Drugs in Reducing Inflammatory and Senescence Burden in Cells from Painful Degenerating IVDs"

_biomolecules, 2023, doi:10.3390/biom13081257_

Round 1

Reviewer 1 Report

The work by Mannarino et al describes how a senolytic combination therapy could prove to be beneficial in the treatment of IVD. The manuscript is well written and easy to understand. But there are issues that need to be addressed before it can be accepted for publication in Biomolecules. 

1. The authors say in the introduction that IVDs have increased expression of TLR 1,2, 4 and 6. But Fig 1C shows no  significant increase except for TLR2 when compared to baseline. Why the  discrepancy?

2. What is the basis for the selection of the concentrations of RG and o-V used in the study? Please provide a rationale for this selection in the methodology section. 

3. Please check statistics for all figures - esp. Fig 1A - Check CCL7 where a statistical significance is shown between the white and red bars!

4. The main issue with this manuscript is the use of a single senescence marker - p16 - that too at the mRNA level. Even though SASP factors are shown they are not exclusive senescence markers as they are regulated by other inflammatory pathways as well. Other senescence markers must be included - p21/p53/lamin B1/SA-b gal, etc. 

5. The inclusion of Caspase 3 as a marker for apoptosis is not convincing. The activated form - Cleaved Caspase 3 is the better representation for apoptosis. This must be shown to demonstrate the effect on apoptosis. 

6. Please check lines 337 - 339. Missing in figure and text?

7. Figure 3G: How does the ratio for p16/Caspase 3 increase for the combination treatment? As the combination treatment has been shown to reduce p16 and increase Caspase 3 cells. Please provide an explanation. 

8. Fig 3: Alamar blue assay. How does it remain unchanged, when the treatment has been shown to induce apoptosis/cell death? 

9. Figure 5B: Is the image from a combination treatment? Please provide images for other treatment conditions too to help appreciate the effect on neurite sprouting between the groups.

Minor comments:

1. Line 149 - "loss of senolytic activity'?

2. Line 153 - four days? Is that correct?

3. Line 349. Is it 50uM o-V? 

4. Fig. 4 heading - Neurotrophines

5. Lines 457, 458 - it is misleading. Please rewrite. 

Author Response

Reviewer 1:

1. The authors say in the introduction that IVDs have increased expression of TLR 1,2, 4 and 6. But Fig 1C shows no significant increase except for TLR2 when compared to baseline. Why the discrepancy?

We thank the reviewer for their comment. The discrepancy that you note relates to the comparators used. In the introduction and according to the cited references Klawitter et al., 2014 (PMID: 24997157); Krock et al., 2017 (PMID: 29215065)), increased expression of TLR 1,2,4 and 6 in IVDs is seen when comparing degenerate IVDs to non-degenerate IVDs. In our model (Figure 1C), we saw an increase in TLR 2 alone when comparing degenerate IVDs induced with Pam2CSK4 as compared to degenerate IVDs.

2. What is the basis for the selection of the concentrations of RG and o-V used in the study? Please provide a rationale for this selection in the methodology section. 

We thank the reviewer for their comment.

The following modification to the methodology section was made on lines 145-149: The concentration of both compounds is based on our previous work and the work of other groups (Cherif et al., 2019; Cherif et al., 2020; Iancu-Rubin et al., 2014; Mannarino et al., 2021; Shah et al., 2019). These studies determined 100 µM of o-Vanillin and 5 µM of RG7112 to be non-toxic and provide the strongest senolytic activity.

To further address your comment about o-Vanillin, we have previously demonstrated the non-toxic concentration range for o-Vanillin. No cytotoxic effect was observed for o-Vanillin between 5–200 µM according to Cherif et al., 2019 (PMID: 30934902) (Figure 3A). Furthermore, in the study by Cherif et al., 2019 (PMID: 30934902), Cherif et al., 2020 (PMID: 32821059) and Shah et al., 2019 (PMID: 30273982), 100 μM of o-Vanillin was found to have senolytic activity on degenerate IVD cells. Higher concentrations did not increase the senolytic activity. For RG-7112, we have previously demonstrated in Cherif et al., 2020 (PMID: 32821059) that RG-7112 at 5µM is not cytotoxic and has senolytic activity on the IVD cells (Figure 1B & Supplementary Figure 1F). Furthermore, previous in vitro studies with RG-7112 have used 5µM in various cell types and models (PMID: 24309210, Unity Patent: US10478432B2). Higher doses did not increase senolytic activity.

3. Please check statistics for all figures - esp. Fig 1A - Check CCL7, where a statistical significance is shown between the white and red bars!

We thank the reviewer for their comment. We have double-checked the statistics for all our graphs. For the CCL7 in Figure 1A specifically, the p-value (p=0.015001) is the same as mentioned in the text on line 273.

4. The main issue with this manuscript is using a single senescence marker - p16 - at the mRNA level. Even though SASP factors are shown, they are not exclusive senescence markers as they are regulated by other inflammatory pathways as well. Other senescence markers must be included - p21/p53/lamin B1/SA-b gal, etc. 

We thank the reviewer for their comment. We agree with the reviewer that there is no perfect marker. We selected p16 as our senescence marker for several reasons over SA-β-gal, p21 and p53.  SA-β-gal will detect both senescent and quiescent cells and so overestimate the number of senescent cells; please see our publication by Cherif et al. 2019 in Figure 2 (PMID: 30934902).

p16 could underestimate the level of senescence, but initial preliminary studies found that p16 is a more reliable marker than p21 and p53 in human IVD cells. We have since used p16 as our senescent marker for three consecutive published studies. (PMID: 30934902, PMID: 32821059, PMID: 33863359). Furthermore, other studies on IVD degeneration have routinely used p16 as their senescence marker (PMID: 30900385, PMID: 34480023).

5. The inclusion of Caspase 3 as a marker for apoptosis is not convincing. The activated form - Cleaved Caspase 3 is the better representation of apoptosis. This must be shown to demonstrate the effect on apoptosis. 

We thank the reviewer for their comment. The methodology used here to detect senolytic activity is based on our previous publications (PMID: 30934902, PMID: 32821059, PMID: 33863359), which investigated the correlation between the increase in Caspase-3 positive cells, decrease in p16 positive, and increase in Ki-67 positive cells as well as the non-cytotoxic effects measured by metabolic activity. We agree that the difference in Caspase-3/7 activity is another valid model for looking at the apoptotic effect of senolytic treatments. We have previously done the Caspase-3/7 in our study (Cherif et al. 2019, Figure 4, PMID: 30934902), and it was seen that the results from IHC of Caspase-3 were consistent with the Caspase 3/7 results. Furthermore, previous literature provides a potential explanation of how senescent cells resist cell death via Caspase-3; this is done by downregulating the expression of Caspase-3, which is responsible for the execution of apoptosis upon the mitochondrial release of cytochrome c and activation of the caspase pathway (PMID: 15541772).

6. Please check lines 337 - 339. Missing in figure and text?

We thank the reviewer for their comment. These graphs were removed from the manuscript prior to submission. These lines have been removed from the figure legends in the manuscript, and the figure legend for Figure #2 has been adapted accordingly.

7. Figure 3G: How does the ratio for p16/Caspase 3 increase for the combination treatment? As the combination treatment has been shown to reduce p16 and increase Caspase 3 cells. Please provide an explanation. 

We thank the reviewer for their comment. The graph in Figure 3G does not represent a ratio of p16 to Caspase-3 but rather the co-localization of p16 and Caspase-3, as highlighted in the text. The heading and the figure legend for the graph have now been changed to highlight better that the figure is discussing the colocalization of p16 & Caspase-3 and not a ratio.

8. Fig 3: Alamar blue assay. How does it remain unchanged when the treatment has been shown to induce apoptosis/cell death? 

We thank the reviewer for their comment. The unchanged metabolic activity can be explained by the increased proliferation (increased Ki-67 positive cells, Figure 3E). This could make up for the increased amount of cell death that is caused by the treatment.

9. Figure 5B: Is the image from a combination treatment? Please provide images for other treatment conditions, too, to help appreciate the effect on neurite sprouting between the groups.

We thank the reviewer for their comment. The image in Figure 5 is a zoomed image from the combination treatment group and was provided to demonstrate how we define a positive and a negative cell. The graph is then generated by counting 10 images from 7 different donors for each condition. This would equal over 2000 cells being quantified for each condition. We can include a supplementary figure with a representative image for each condition if the reviewer thinks it would add to the manuscript. 

Minor comments:

1. Line 149 - "loss of senolytic activity'?

We thank the reviewer for their comment.

Edit to line 142:  Serial dilutions with 0.25/0.5/2.5/5 μM of RG-7112 (Selleck Chemicals, Houston, TX, USA) or 5/10/50/100 μM of o-Vanillin (Sigma-Aldrich, Oakville, ON, Canada) was performed to identify the lowest concentrations with senolytic activity.

2. Line 153 - four days? Is that correct?

We thank the reviewer for their comment. Yes, four days of senolytic treatments are correct; as further shown in the timeline in Figure 3A, senolytic treatment was from Day 4 to Day 8.

3. Line 349. Is it 50uM o-V? 

We thank the reviewer for their comment. The change to line 349 has been made.

Edit to line 349: Moreover, 2.5μM of RG-7112 and 50 μM of o-Vanillin were further investigated, given that they both resulted in a significant decrease in the expression of p16ink4a. Four combinations were assessed, 5 μM of RG-7112 and 100 μM of o-Vanillin; 2.5 μM of RG-7112 and 50 μM of o-Vanillin.

4. Fig. 4 heading – Neurotrophins

We thank the reviewer for their comment. We have changed the heading in Figure 4 to Neurotrophins.

5. Lines 457, 458 - it is misleading. Please rewrite. 

We thank the reviewer for their comment, lines 457 and 458 have been changed. 

Reviewer 2 Report

Dear Authors:  Thank you for providing your manuscript detailing the possible beneficial effects of senolytic combination treatment to address inflammatory and senescence burden in cells from painful degenerating IVDs.  There are a number of areas where I struggle with this manuscript.

1.    The methods section clearly states that you used a validated 3D cell pellet culture system to mimic the IVD milieu followed by several self-citations from your group's published work.  Pellet cultures are far more commonly used to emulate the cartilage and do not represent the 3-D milieu of the IVD (Two- and three-dimensional in vitro nucleus pulposus cultures.  Please see "An in silico analysis of local nutrient microenvironments", McDonnell EE and Buckley CT, JOR Spine. 2022;5:e1222. jorspine.com https://doi.org/10.1002/jsp2.1222.  Within this manuscript it is specifically stated "...Lastly, when investigating cell pellet culture and smaller microaggregate systems, it is evident that the conventional pellet configuration of 250,000 cells is not a good representative of the IVD nutrient microenvironment. The model of pellet culture was derived from the cartilage field with high cell numbers, low oxygen conditions and supraphysiological levels of glucose. Although the disc field appears to gravitate toward NX conditions, the current work highlights that despite the high external boundary concentration, the local cellular oxygen is still predicted to reach HX conditions which are not comparable to in vivo disc oxygen measurements."   Why did you choose a pellet culture method as opposed to the more standardized alginate bead or agarose 3D culture methods?

Another manuscript cited within this paper that was ostensibly meant to support the use of senolytic treatment ("Long-term treatment with senolytic drugs Dasatinib and Quercetin ameliorates age dependent intervertebral disc degeneration in mice.  In this paper the authors report on their results of weekly intraperitoneal injection of Senolytic agents with respect to the IVD)", Novais EJ, et al (2021) 12:5213 | https://doi.org/10.1038/s41467-021-25453-2.  Within this paper the authors state that their intervention had no effect upon already degenerating IVDs and curiously have no effect upon OA affecting the knee.  Since the use of these drugs had no effect upon DDD already in process, and the current study is limited only to in vitro methods (and a debatable 3D model), how do the current authors think that their proposal of multiple Senolytic agents might modify DDD already in process?

I am puzzled by the claims that the Senolytic agents used in this study had any impact upon cell death/apoptosis.  You used semi-quantiative immunocytochemical methods to probe for the expression of Casapase-3, the 'executioner' Caspase in this cell death sequence of intracellular apoptotic signaling.  However, the presence of Caspase-3 does not indicate apoptosis.  Rather the phase of Caspase signaling that marks the irreversible step towards cell death occurs with cleavage of Caspase-3 to 'activated 'Caspase -3(-7)'.  Why did you not use Western blotting or so other form (FACS?) to evaluate for activated Caspase-3?  I struggle to accept any data concerning apoptosis from what I have seen in your manuscript UNLESS the immunocytochemical analysis you describe specifically recognizes the active/cleaved form of Caspase-3, please clarify?

Finally, and on a translational level, given the cited prior report that these Senolytic agents do not affect DDD that is already underway AND the paper indicating some effect in mice IVDs required weekly intraperitoneal injections, how do you anticipate that a therapeutic approach like that presented here might be translated to a clinical application?  

Author Response

Reviewer #2

1.    The methods section clearly states that you used a validated 3D cell pellet culture system to mimic the IVD milieu, followed by several self-citations from your group's published work.  Pellet cultures are far more commonly used to emulate the cartilage and do not represent the 3-D milieu of the IVD (Two- and three-dimensional in vitro nucleus pulposus cultures.  Please see "An in silico analysis of local nutrient microenvironments", McDonnell EE and Buckley CT, JOR Spine. 2022;5:e1222. jorspine.com https://doi.org/10.1002/jsp2.1222.  Within this manuscript, it is specifically stated, "...Lastly, when investigating cell pellet culture and smaller microaggregate systems, it is evident that the conventional pellet configuration of 250,000 cells is not a good representative of the IVD nutrient microenvironment. The model of pellet culture was derived from the cartilage field with high cell numbers, low oxygen conditions and supraphysiological levels of glucose. Although the disc field appears to gravitate toward NX conditions, the current work highlights that despite the high external boundary concentration, the local cellular oxygen is still predicted to reach HX conditions which are not comparable to in vivo disc oxygen measurements."   Why did you choose a pellet culture method as opposed to the more standardized alginate bead or agarose 3D culture methods?

We thank the reviewer for their comment. The study by McDonnell et al. describes well the importance of various parameters when conducting a 2D and 3D culture model. We understand that one of the comments in this study, as stated by the reviewer, is that the pellet configuration with 250,000 cells is not the most optimal representation of an IVD microenvironment. Unfortunately, this study was published in 2022, and by this point, our experiment had largely been completed. More importantly, we work with human IVDs from low back pain patients; given the scarcity and limited availability of samples, obtaining a larger quantity of cells to produce larger pellets would translate into having to use a higher passage of the cells (i.e. cells with lost phenotype) or a lower n per group which we feel would impact more than having smaller pellets.

Regarding the model, we are surprised by the reviewer’s comment that our current pellet model is not representative of a real disc. Our previous study by Cherif et al. 2020 tested both o-Vanillin and RG-7112 in the same pellet model as the current study and ex vivo intact disc model (i.e., the ideal model). In this study, it was found that the results are comparable between the intact disc and pellet model when observing the senolytic activity of both O-Vanillin and RG-7112. Furthermore, we tested the single drugs in this model, and we observed the senolytic effect in this model in three publications in journals of high stature in the field (PMID: 30934902, PMID: 32821059, PMID: 33863359). To remain consistent and to test the combination of the same compounds, we used the same culture model. Moving forward, we will further validate our findings in both an ex vivo organ culture model and in vivo mouse model. Moreover, we decided against using alginate or agarose since we did not want to change our current functional model, and we were not sure if the alginate or agarose would have any interaction with our compounds or induce auto-fluorescence in our staining or other caveats that are not present in our current model.

2. Another manuscript cited within this paper that was ostensibly meant to support the use of senolytic treatment ("Long-term treatment with senolytic drugs Dasatinib and Quercetin ameliorates age dependent intervertebral disc degeneration in mice.  In this paper the authors report on their results of weekly intraperitoneal injection of Senolytic agents with respect to the IVD)", Novais EJ, et al (2021) 12:5213 | https://doi.org/10.1038/s41467-021-25453-2.  Within this paper the authors state that their intervention had no effect upon already degenerating IVDs and curiously have no effect upon OA affecting the knee.  Since the use of these drugs had no effect upon DDD already in process, and the current study is limited only to in vitro methods (and a debatable 3D model), how do the current authors think that their proposal of multiple Senolytic agents might modify DDD already in process?

We thank the reviewer for their comment. The findings from Novais et al. 2021, concluded that “the potential benefits of D + Q treatment include alleviation of disc degeneration, reduction in systemic inflammation, and improved physical condition during aging.” The reviewers' comments are only addressing the Novais et al. older cohort, but in the middle-aged mild degenerate cohort, the authors were able to see similar findings to our current study regarding the use of combining senolytic compounds. In addition, o-Vanillin and RG-7112 are not targeting the same anti-apoptotic and pro-senescent pathways as D + Q. Furthermore, as we have mentioned in the comments above regarding the potential of o-Vanillin and RG-7112, we have already published our findings in three prior publications in journals of good stature (PMID: 30934902, PMID: 32821059, PMID: 33863359). There are many advantages for combination therapy: 1) concurrently targeting multiple and indirectly related anti-apoptotic pathways may result in increased selectivity for senescent cells in the absence of toxicity for normal proliferating or quiescent cells, 2) targeting antiapoptotic networks instead of a single target and 3) therapeutic dosages can potentially be lowered by combinations all the while decreasing side effects associated with single drugs. To further support our work, a Clinical Phase II trial initiated in 2020 with a combination of Fisetin, Dasatinib, and Quercetin is currently in progress. Moreover, in cancer treatment, combining chemotherapy with metformin, a known inhibitor of the SASP, provides prolonged tumour remission. These are only some examples of why we believe that the use of combination therapy could have a potential benefit in IVD degeneration and further supports our work. 

3. I am puzzled by the claims that the Senolytic agents used in this study had any impact upon cell death/apoptosis.  You used semi-quantiative immunocytochemical methods to probe for the expression of Casapase-3, the 'executioner' Caspase in this cell death sequence of intracellular apoptotic signaling.  However, the presence of Caspase-3 does not indicate apoptosis.  Rather the phase of Caspase signaling that marks the irreversible step towards cell death occurs with cleavage of Caspase-3 to 'activated 'Caspase -3(-7)'.  Why did you not use Western blotting or so other form (FACS?) to evaluate for activated Caspase-3?  I struggle to accept any data concerning apoptosis from what I have seen in your manuscript UNLESS the immunocytochemical analysis you describe specifically recognizes the active/cleaved form of Caspase-3, please clarify?

We thank the reviewer for their comment. The methodology used here to detect senolytic activity is based on our previous publications (PMID: 30934902, PMID: 32821059, PMID: 33863359), which investigated the correlation between the increase in Caspase-3 positive cells, decrease in p16 positive, and increase in Ki-67 positive cells as well as the non-cytotoxic effects measured by metabolic activity. We agree that the difference in Caspase-3/7 activity is another valid model for looking at the apoptotic effect of senolytic treatments. We have previously done the Caspase-3/7 in our study (Cherif et al. 2019, Figure 4, PMID: 30934902), and it was seen that the results from IHC of Caspase-3 were consistent with the Caspase 3/7 results. Furthermore, previous literature provides a potential explanation of how senescent cells resist cell death via Caspase-3; this is done by downregulating the expression of Caspase-3, which is responsible for the execution of apoptosis upon the mitochondrial release of cytochrome c and activation of the caspase pathway (PMID: 15541772).

4. Finally, and on a translational level, given the cited prior report that these Senolytic agents do not affect DDD that is already underway AND the paper indicating some effect in mice IVDs required weekly intraperitoneal injections, how do you anticipate that a therapeutic approach like that presented here might be translated to a clinical application?

We thank the reviewer for their comment. Excitingly there are already senolytic treatments that have been translated into clinical applications for a variety of diseases. One example is the current clinical trial of intermittent Dasatinib and Quercetin(Dasatinib:100 mg/day, Quercetin:1250 mg/day, three days/week over three weeks) was conducted in participants with idiopathic pulmonary fibrosis (n = 14) to evaluate the feasibility of implementing a senolytic intervention. Initial evidence shows that senolytics may alleviate physical dysfunction in idiopathic pulmonary fibrosis, warranting the evaluation of Dasatinib and Quercetin in larger randomized controlled trials for senescence-related diseases (PMID: 30616998). Moreover, a clinical trial initiated in 2020 with a combination of Fisetin, Dasatinib and Quercetin, which aims to determine if senolytic drugs reduce senescent cell burden and reduce bone resorption markers/increase bone formation markers in elderly women, is also underway. These studies were all based on in vitro studies which have now been scaled up to clinical studies and hopefully will be implemented in practice. As such, they provide promising avenues for our senolytic treatments like o-Vanillin and RG-7112. Novais et al. 2021 did not present any results indicating that the effect was lost or reduced with fewer than weekly injections. They also didn’t present data using other administration methods. It is clear that administration and frequency must be carefully investigated in humans. An encouraging finding from our publication by Cherif et al. 2020 (PMID: 32821059) is that a single dose of either drug alone was sufficient to significantly reduce the number of senescent cells and SASP factor release, while proteoglycan content was increased in intact human IVDs. This can be explained by the hit-and-run theory based on the fact that when the senescent cells are removed, the SASP is also removed. Removing the SASP will prevent SASP amplification and further induction of senescence.

Reviewer 3 Report

Seeking new senolytic drugs is a hot topic in biogerontological studies, therefore the aim of the study is very timely. This manuscript is a continuation of previous research of this group. They published earlier data showing that RG-7112 and o-Vanilin are promising senolytic drugs that would be helpful in the reduction of low back pain related to intervertebral disc (IVD) degeneration. Now they have shown that the combination of these two compounds is more potent than a single treatment. It is an element of novelty. The manuscript is written clearly and well-illustrated, the results are well-elaborated from the statistical point of view, and their interpretation is convincing. The results support the conclusions. The work requires minor corrections and explanations. Once completed, it will meet the criteria required for publication in Biomolecules.

Detailed comments:

·        It is written: “The FDA-approved senolytic compound RG-7112 (RO5045337) (Weber, 2010) restores p53 physiological activity via MDM2 (Tovar et al., 2013; Weber, 2010). RG-7112 has been shown to specifically kill senescent IVD cells (Cherif et al., 2020; Mannarino et al., 2021)”. The readers can get the impression that the FDA approved this compound as a senolytic drug. It will be better to write: The FDA-approved compound RG-7112 (RO5045337) (Weber, 2010) restores p53 physiological activity via MDM2 (Tovar et al., 2013; Weber, 2010).

·        The Authors have written: “Serial dilutions with 0.25/0.5/2.5/5 μM of RG-7112 (Selleck Chemicals, Houston, TX, USA) or 5/10/50/100 μM of o-Vanillin (Sigma-Aldrich, Oakville, ON, Canada) was performed to identify concentrations which do not demonstrate result in loss of senolytic activity.” The Authors were probably looking for the most effective senolytic effect by the lowest concentration. In such manner it was written in line 311: „Identifying the lowest effective concentration of RG-7112 and o-Vanillin at which senolytic activity is preserved in monolayer culture”.

·        It has been written in lines 166-168: “Rat adrenal pheochromocytoma (PC12) cell line ……... They are commonly used to study neuronal differentiation and neurite sprouting 20-22”. There is a mistake. What does it mean 20-22?

·        Fig. 2, above the pictures summarizing data concerning Ki67, p16, caspase 3 it is written "serial dilution". It is better to remove the phrase “serial dilution” as it's a bit confusing. One can get the impression that cells were treated with different concentrations of antibodies/proteins of Ki67, p16, caspase 3, and not that different amounts of the analyzed proteins were detected following administration of different senolytic mixture concentrations.

·        Line 349: “Given that 5μM of RG-7112 and 100μM of o-Vanillin resulted in the most significant decrease in senescence, combinations of these concentrations were selected for future experiments. Moreover, 2.5μM of RG-7112 and 50 μM of o-Vanillin were further investigated, given that they both resulted in a significant decrease in the expression of p16ink4a. Four combinations were assessed, 5 μM of RG-7112 and 100 μM of o-Vanillin; 2.5 μM of RG-7112 and 50 μM of o-Vanillin”. Only two combinations are mentioned. Moreover, in Fig. 3 there are 3 combinations plus RG-7112 or o-Vanillin alone. In addition, the concentrations given in the text are inconsistent with those in the graphs. The combination 2.5μM of RG-7112 and 50μM of o-Vanillin, which are mentioned in the description, is missing in the figure.

·        Fig. 3 - Remove the word “combination” in D, E, F because it is confusing and add the word “colocalization” instead of slash (/) in the figure description above the graphs concerning p16 and caspase 3.

·         Line 417: in Fig. 3 description “(f-j) Magnified images of (a-c)” should be a-e

·        Analysis of levels of neurite sprouting and neurite growth. While the decrease in gene expression after the combination of senolytics is significant, the ability to form sprouts/neurites is presented less convincingly. A positive control (treatment with TLR-2/6 agonist) and senolytic post-treatment image should be shown.

·        In the discussion it would be helpful to include the aspect of high concentrations of the compounds used and address potential limitations of their practical use.

Author Response

Reviewer # 3:

Seeking new senolytic drugs is a hot topic in biogerontological studies, therefore the aim of the study is very timely. This manuscript is a continuation of previous research of this group. They published earlier data showing that RG-7112 and o-Vanilin are promising senolytic drugs that would be helpful in the reduction of low back pain related to intervertebral disc (IVD) degeneration. Now they have shown that the combination of these two compounds is more potent than a single treatment. It is an element of novelty. The manuscript is written clearly and well-illustrated, the results are well-elaborated from the statistical point of view, and their interpretation is convincing. The results support the conclusions. The work requires minor corrections and explanations. Once completed, it will meet the criteria required for publication in Biomolecules.

Detailed comments:

  • It is written: “The FDA-approved senolytic compound RG-7112 (RO5045337) (Weber, 2010) restores p53 physiological activity via MDM2 (Tovar et al., 2013; Weber, 2010). RG-7112 has been shown to specifically kill senescent IVD cells (Cherif et al., 2020; Mannarino et al., 2021)”. The readers can get the impression that the FDA approved this compound as a senolytic drug. It will be better to write: The FDA-approved compound RG-7112 (RO5045337) (Weber, 2010) restores p53 physiological activity via MDM2 (Tovar et al., 2013; Weber, 2010). 

We thank the reviewer for their comment. We have made the change to line 84: The FDA-approved compound RG-7112 (RO5045337)(Weber, 2010) restores p53 physiological activity via MDM2 (Tovar et al., 2013; Weber, 2010).

  • The Authors have written: “Serial dilutions with 0.25/0.5/2.5/5 μM of RG-7112 (Selleck Chemicals, Houston, TX, USA) or 5/10/50/100 μM of o-Vanillin (Sigma-Aldrich, Oakville, ON, Canada) was performed to identify concentrations which do not demonstrate result in loss of senolytic activity.” The Authors were probably looking for the most effective senolytic effect by the lowest concentration. In such manner it was written in line 311: „Identifying the lowest effective concentration of RG-7112 and o-Vanillin at which senolytic activity is preserved in monolayer culture”.

We like to thank the reviewer for their comment. The change has been made to lines 142-143: Serial dilutions with 0.25/0.5/2.5/5 μM of RG-7112 (Selleck Chemicals, Houston, TX, USA) or 5/10/50/100 μM of o-Vanillin (Sigma-Aldrich, Oakville, ON, Canada) was performed to identify concentrations that are the most potent andthe lowest concentration of each senolytic.

  • It has been written in lines 166-168: “Rat adrenal pheochromocytoma (PC12) cell line ……... They are commonly used to study neuronal differentiation and neurite sprouting 20-22”. There is a mistake. What does it mean 20-22?

We thank the reviewer for their comment. We apologize for the error. A reference was supposed to be put there, we have added the appropriate reference on lines 171-172: When exposed to NGF, they take on a neuronal-like phenotype. They are commonly used to study neuronal differentiation and neurite sprouting (Greene et al., 1976 and Krock et al., 2014).

  • Fig. 2, above the pictures summarizing data concerning Ki67, p16, caspase 3 it is written "serial dilution". It is better to remove the phrase “serial dilution” as it's a bit confusing. One can get the impression that cells were treated with different concentrations of antibodies/proteins of Ki67, p16, caspase 3, and not that different amounts of the analyzed proteins were detected following administration of different senolytic mixture concentrations.

We like to thank the reviewer for their comment. Fig. 2 has been adapted.

  • Line 349: “Given that 5μM of RG-7112 and 100μM of o-Vanillin resulted in the most significant decrease in senescence, combinations of these concentrations were selected for future experiments. Moreover, 2.5μM of RG-7112 and 50 μM of o-Vanillin were further investigated, given that they both resulted in a significant decrease in the expression of p16ink4a. Four combinations were assessed, 5 μM of RG-7112 and 100 μM of o-Vanillin; 2.5 μM of RG-7112 and 50 μM of o-Vanillin”. Only two combinations are mentioned. Moreover, in Fig. 3 there are 3 combinations plus RG-7112 or o-Vanillin alone. In addition, the concentrations given in the text are inconsistent with those in the graphs. The combination 2.5μM of RG-7112 and 50μM of o-Vanillin, which are mentioned in the description, is missing in the figure.

We thank the reviewer for their comment. We have made the changes to the text to better describe the three combinations that are shown in the Figure on lines 352-353: “Three combinations were assessed; 5 μM of RG-7112 and 100 μM of o-Vanillin; 2.5 μM of RG-7112 and 100 μM of o-Vanillin, and 5 μM of RG-7112 and 50 μM of o-Vanillin”.

  • Fig. 3 - Remove the word “combination” in D, E, F because it is confusing and add the word “colocalization” instead of slash (/) in the figure description above the graphs concerning p16 and caspase 3.

We like to thank the reviewer for their comment. We have adapted Fig 3 accordingly.

  • Line 417: in Fig. 3 description “(f-j) Magnified images of (a-c)” should be a-e

We thank the reviewer for their comment. We have made the correction to the figure legend on line 417.

  • Analysis of levels of neurite sprouting and neurite growth. While the decrease in gene expression after the combination of senolytics is significant, the ability to form sprouts/neurites is presented less convincingly. A positive control (treatment with TLR-2/6 agonist) and senolytic post-treatment image should be shown. 

We thank the reviewer for their comment. The PC-12 representative images are now presented in Supplementary Figure 1. We apologize for the lack of clarity, image A (a) is in fact a representative image for sprouting with only TLR-2/6 agonist.

  • In the discussion, it would be helpful to include the aspect of high concentrations of the compounds used and address potential limitations of their practical use.

We thank the reviewer for the comment we have added to the discussion regarding the limitation of high concentrations on lines 538-543: Previous studies have demonstrated that lower concentrations of RG-7112, like those used here, are safe and beneficial in musculoskeletal diseases, whereas, at the high concentrations, used for hematological cancer therapy, toxic side effects were described(Hassin & Oren, 2023). As for, o-Vanillin, the safety of the compound was evaluated to have an IC50=4.2 mM, but no higher concentration than 100μM was tested.

Reviewer 4 Report

This study used TLR2 to create a senescent and inflammation cell model, and studied if combining the two senolytic drugs, o-Vanillin and RG-7112, could efficiently remove senescent cells, and reduce the release of inflammatory factors and pain. It is found that compared to the single treatments, the combination of o-Vanillin and RG-7112 significantly reduced the amount of senescent IVD cells, pro-inflammatory cytokines and neurotrophic factors. Moreover, both single and combination treatments significantly reduced neuronal sprouting in PC-12 cells.

I have a few questions for the authors:

1.Why the authors submitted a manuscript with revision marks and changes tracked. This is not the usual practice.

2. The full names were not provided when the abbreviations first appeared, such as for TLR2.

3. Many experiments, such as those in Figure1A and Figure3G/H, only have the modeling group and the treatment group. It is recommended to add a healthy control group that does not recieve any medication. 

4. Figure2: How to count the positivity of p16, Ki67 or Caspase 3? It seems to me that all cells are positive for these three markers. So here comes the question: Are only the cells with positive nucleus counted as positive cells? 

5. Figure3B and 3C immunofluorescence study: It is unclear at what time point were the samples of these images collected, which should be specified. More importantly, only the staining photos of a single sample were provided in Figure3B and 3C. Original images should be provided for each group, at least one image per group. 

6. Figure5: Is this an experiment to prove that these two molecules can reduce pain? If so, it is recommended to add animal tests for pain measurement.

Author Response

Reviewer #4:

This study used TLR2 to create a senescent and inflammation cell model, and studied if combining the two senolytic drugs, o-Vanillin and RG-7112, could efficiently remove senescent cells, and reduce the release of inflammatory factors and pain. It is found that compared to the single treatments, the combination of o-Vanillin and RG-7112 significantly reduced the amount of senescent IVD cells, pro-inflammatory cytokines and neurotrophic factors. Moreover, both single and combination treatments significantly reduced neuronal sprouting in PC-12 cells.

I have a few questions for the authors:

  1. Why the authors submitted a manuscript with revision marks and changes tracked. This is not the usual practice.

We apologize to the reviewer that the track changes were in the manuscript. This is because this is our second re-submission, and the manuscript was with revision marks and changes tracked, following the editor's instructions to allow previous reviewers to see the changes made.

  1. The full names were not provided when the abbreviations first appeared, such as for TLR2.

We like to thank the reviewer for their comment we have now addressed all first appearances of abbreviations in the manuscript.

  1. Many experiments, such as those in Figure1A and Figure3G/H, only have the modeling group and the treatment group. It is recommended to add a healthy control group that does not recieve any medication

We thank the reviewer for their comment. For Figure 1A, the data is normalized to the baseline control, meaning the horizontal line at 1 represents the baseline control. Regarding Figure 3 G/H we chose not to include the baseline control to avoid confusion regarding the control (TLR2/6 stimulated) and baseline.

  1. Figure2: How to count the positivity of p16, Ki67 or Caspase 3? It seems to me that all cells are positive for these three markers. So here comes the question: Are only the cells with positive nucleus counted as positive cells? 

Regarding the p16INK4a, Caspase 3 and Ki-67 detection, only cells with nuclear immunostaining were considered positive. This has now been clarified in the text in lines 374-375. We and others have used the same method in previous publications that are now referred to. (Cherif et al,2019 and 2020; Liu et al, 2007; Shidham et al, 2011).

  1. Figure3B and 3C immunofluorescence study: It is unclear at what time point were the samples of these images collected, which should be specified. More importantly, only the staining photos of a single sample were provided in Figure3B and 3C. Original images should be provided for each group, at least one image per group. 

We thank the reviewer for their comment. The images were taken at the end of the culture (day 24). This statement has now been added to the methods section on line 163: After Day 24, cell pellets were washed in PBS, and fixed with 4% paraformaldehyde. We have prepared supplemental figures 1,2, and 3 with one image per condition.

  1. Figure 5: Is this an experiment to prove that these two molecules can reduce pain? If so, it is recommended to add animal tests for pain measurement.

We thank the reviewer for the suggestion. The goal of this study was to evaluate if the combination treatment would improve the senolytic and anti-inflammatory potential of single-drug treatments. As the reviewer pointed out, this type of experiment doesn’t prove that the drugs reduce pain, it is showing the potential, but this must be validated in in vivo experiments. This will be evaluated in the future by our group and others in models with higher translational value and in similar pathological contexts.

Round 2

Reviewer 1 Report

Please address Query 5 and 9.

Author Response

We thank the reviewer for the comments and hope that our responses are satisfactory. 

  1. The inclusion of Caspase 3 as a marker for apoptosis is not convincing. The activated form - Cleaved Caspase 3 is the better representation of apoptosis. This must be shown to demonstrate the effect on apoptosis.

We apologize for the lack of clarity regarding the antibody we used. We used C8487, from Sigma-Aldrich recognizing cleaved caspase-3.  It is generated to the cleavage site 17-19kDa and purified on un-cleaved Caspase to remove any Ab’s recognizing the intact form 32kDa. The antibody number has been added to the methods section and we clarified in the result section that it is recognizing cleaved Caspase 3 

  1. Figure 5B: Is the image from a combination treatment? Please provide images for other treatment conditions, too, to help appreciate the effect on neurite sprouting between the groups.

We have now added a supplementary figure (Supplementary Figure 1) showing lower magnification images in response to all treatments.

Reviewer 2 Report

Dear authors:  Thank you for providing your replies to my questions.  Unfortunately, although you do thank me for my observations, largely you ignore my comments (see below).

We thank the reviewer for their comment. The study by McDonnell et al. describes well the importance of various parameters when conducting a 2D and 3D culture model. We understand that one of the comments in this study, as stated by the reviewer, is that the pellet configuration with 250,000 cells is not the most optimal representation of an IVD microenvironment. Unfortunately, this study was published in 2022, and by this point, our experiment had largely been completedIt is not at all novel that 3D the tissue culture of IVD NP cells is inappropriate in pellet format (BMC Musculoskeletal Disorders 2000 Oct 23. doi: 10.1186/1471-2474-1-1 Gruber and Hanley).  Furthermore as reported in "Intervertebral Disc Cells Exhibit Differences in Gene Expression in Alginate and Monolayer Culture, 2001 SPINE Volume 26, Number 16, pp 1747–1752, Wang et al", NP cells can be expanded in monolayer and re-suspended in 3D culture and the cells regain their phenotype.  Therefore the excuse that the originally cited paper comes across more as an excuse than a logical reason.  More importantly, we work with human IVDs from low back pain patients; given the scarcity and limited availability of samples, obtaining a larger quantity of cells to produce larger pellets would translate into having to use a higher passage of the cells (i.e. cells with lost phenotype) or a lower n per group which we feel would impact more than having smaller pellets.

Regarding the model, we are surprised by the reviewer’s comment that our current pellet model is not representative of a real disc. Our previous study by Cherif et al. 2020 tested both o-Vanillin and RG-7112 in the same pellet model as the current study and ex vivo intact disc model (i.e., the ideal model). In this study, it was found that the results are comparable between the intact disc and pellet model when observing the senolytic activity of both O-Vanillin and RG-7112. Furthermore, we tested the single drugs in this model, and we observed the senolytic effect in this model in three publications in journals of high stature in the field (PMID: 30934902, PMID: 32821059, PMID: 33863359). To remain consistent and to test the combination of the same compounds, we used the same culture model. Moving forward, we will further validate our findings in both an ex vivo organ culture model and in vivo mouse model. Moreover, we decided against using alginate or agarose since we did not want to change our current functional model, and we were not sure if the alginate or agarose would have any interaction with our compounds or induce auto-fluorescence in our staining or other caveats that are not present in our current model.  NP cells have been cultured in alginate and immunohistochemically stained in other papers.  Again, this answer comes across more as an excuse than a legitimate scientific explanation.  

With respect to your answer regarding response #2  "....To further support our work, a Clinical Phase II trial initiated in 2020 with a combination of Fisetin, Dasatinib, and Quercetin is currently in progress. Moreover, in cancer treatment, combining chemotherapy with metformin, a known inhibitor of the SASP, provides prolonged tumour remission. These are only some examples of why we believe that the use of combination therapy could have a potential benefit in IVD degeneration and further supports our work."  This is a systemic therapy, treating the IVD via injection is an entirely different story as would the challenges in a human of weekly IP injections in the hope that there would be any effect?  Again, a disappointing and unscientific reply. 

Author Response

Dear Reviewer 2, we did not intend to ignore your review but it was not easy to understand what you wanted us to change. We were given 10 days to submit our revised manuscript which would not allow us to redo the study using a different model. I have kept the text from your review as we received it and indicated in blue the new responses.

1) We thank the reviewer for their comment. The study by McDonnell et al. describes well the importance of various parameters when conducting a 2D and 3D culture model. We understand that one of the comments in this study, as stated by the reviewer, is that the pellet configuration with 250,000 cells is not the most optimal representation of an IVD microenvironment. Unfortunately, this study was published in 2022, and by this point, our experiment had largely been completed.  It is not at all novel that 3D the tissue culture of IVD NP cells is inappropriate in pellet format (BMC Musculoskeletal Disorders 2000 Oct 23. doi: 10.1186/1471-2474-1-1 Gruber and Hanley).  Furthermore as reported in "Intervertebral Disc Cells Exhibit Differences in Gene Expression in Alginate and Monolayer Culture, 2001 SPINE Volume 26, Number 16, pp 1747–1752, Wang et al", NP cells can be expanded in monolayer and re-suspended in 3D culture and the cells regain their phenotype.  Therefore the excuse that the originally cited paper comes across more as an excuse than a logical reason.  More importantly, we work with human IVDs from low back pain patients; given the scarcity and limited availability of samples, obtaining a larger quantity of cells to produce larger pellets would translate into having to use a higher passage of the cells (i.e. cells with lost phenotype) or a lower n per group which we feel would impact more than having smaller pellets.

It was difficult to understand what the reviewer wants us to address, as we could understand we were asked for an explanation as to why we used the model we did. We provided our explanation and rationale for this, and I have expanded on the same below. Did the reviewer ask for an explanation or for the entire study to be redone in another model system and if so on what bases? 

We are not claiming that that model is perfectly mimicking the human in vivo situation. No in vitro model is fully recapitulating the in vivo model. It is clear that model systems will have advantages and disadvantages. The model that is most suitable and feasible must therefore be selected while keeping in mind that it is a model system. The only fully reproducible system would be in humans or as a second-best intact human IVDs. In fact, we have tested the drugs and verified that they act in the same way in intact human IVDs and in monolayer and pellet cultures. Human in vivo studies have not yet been done.  For our study, it was important to expand and culture the cells for an as short period as possible.  This is for several reasons. 1) we are studying senescent cells and it is important to keep as many of the senescent cells from the tissue as possible in our experiments. Expansion will select for highly proliferative cells and that is not what we want to study. However, less expansion means fewer cells and we are forced to use a lower cell number. We are also using TLR activation to boost senescence and IVD cells in our hands lose TLR2 expression passed passage 3, we therefore use the cells in passages 1-2.    2) Alginate cultures are great when you want to study matrix production and regenerative approaches but in our case, it is not the optimal system.

2) Regarding the model, we are surprised by the reviewer’s comment that our current pellet model is not representative of a real disc. Our previous study by Cherif et al. 2020 tested both o-Vanillin and RG-7112 in the same pellet model as the current study and ex vivo intact disc model (i.e., the ideal model). In this study, it was found that the results are comparable between the intact disc and pellet model when observing the senolytic activity of both O-Vanillin and RG-7112. Furthermore, we tested the single drugs in this model, and we observed the senolytic effect in this model in three publications in journals of high stature in the field (PMID: 30934902, PMID: 32821059, PMID: 33863359). To remain consistent and to test the combination of the same compounds, we used the same culture model. Moving forward, we will further validate our findings in both an ex vivo organ culture model and in vivo mouse model. Moreover, we decided against using alginate or agarose since we did not want to change our current functional model, and we were not sure if the alginate or agarose would have any interaction with our compounds or induce auto-fluorescence in our staining or other caveats that are not present in our current model.  NP cells have been cultured in alginate and immunohistochemically stained in other papers.  Again, this answer comes across more as an excuse than a legitimate scientific explanation.  

With respect to your answer regarding response #2  "....To further support our work, a Clinical Phase II trial initiated in 2020 with a combination of Fisetin, Dasatinib, and Quercetin is currently in progress. Moreover, in cancer treatment, combining chemotherapy with metformin, a known inhibitor of the SASP, provides prolonged tumour remission. These are only some examples of why we believe that the use of combination therapy could have a potential benefit in IVD degeneration and further supports our work."  This is a systemic therapy, treating the IVD via injection is an entirely different story as would the challenges in a human of weekly IP injections in the hope that there would be any effect?  Again, a disappointing and unscientific reply. 

Here we tried to clarify why we referred to Novais EJ, et al (2021) 12:5213 | https://doi.org/10.1038/s41467-021-25453-2 and why senolytic therapy could be a potential treatment we never suggested doing weekly injections in human patients. The citation is not vital to our study, but it is published in a highly regarded journal (Nature Communications) and was added as support for senolytic therapy. This study was done in aging mice and is providing support that this type of therapy (not drug administration method) potentially could be translated to human patients.  How treatment in humans would be done is a completely different story. Our study is simply providing evidence that the combination of RG7112 and o-vanillin is removing significantly more senescent human IVD cells and downstream effects than either drug alone in vitro.  Translation to human patients would likely require a delivery system with sustained release. Did the reviewer want us to remove the citation or elaborate on how treatment can be translated to human patients with low back pain?